# A Neurodevelopment Approach for a Transitional Model of Early Onset Schizophrenia

**DOI:** 10.3390/brainsci11020275

**Published:** 2021-02-23

**Authors:** Domenico De Berardis, Sergio De Filippis, Gabriele Masi, Stefano Vicari, Alessandro Zuddas

**Affiliations:** 1Department of Mental Health, Psychiatric Service of Diagnosis and Treatment, Hospital “G. Mazzini,” National Health Service (NHS), 64100 ASL 4 Teramo, Italy; 2Department of Neurosciences and Imaging, University “G. D’Annunzio”, 66100 Chieti, Italy; 3Department of Neuropsychiatry, Villa von Siebenthal Neuropsychiatric Hospital and Clinic, Genzano di Roma, 100045 Rome, Italy; sergio.defilippis@me.com; 4IRCCS Stella Maris, Scientific Institute of Child Neurology and Psychiatry, Calambrone, 56128 Pisa, Italy; gabriele.masi@fsm.unipi.it; 5Department of Life Sciences and Publich Health, Catholic University, 00135 Rome, Italy; stefano.vicari@opbg.net; 6Child & Adolescent Psychiatry, Bambino Gesù Children’s Hospital, 00168 Rome, Italy; 7Child and Adolescent Neuropsychiatry Unit, Department of Biomedical Sciences, University of Cagliari and “A Cao” Paediatric Hospital, “G Brotzu” Hospital Trust, 109134 Cagliari, Italy; azuddas.unica@gmail.com

**Keywords:** transitional psychiatry, schizophrenia, autism, ADHD

## Abstract

In the last decades, the conceptualization of schizophrenia has dramatically changed, moving from a neurodegenerative process occurring in early adult life to a neurodevelopmental disorder starting be-fore birth, showing a variety of premorbid and prodromal symptoms and, in relatively few cases, evolving in the full-blown psychotic syndrome. High rates of co-occurring different neurodevelopmental disorders such as Autism spectrum disorder and ADHD, predating the onset of SCZ, and neurobio-logical underpinning with significant similarities, support the notion of a pan-developmental disturbance consisting of impairments in neuromotor, receptive language, social and cognitive development. Con-sidering that many SCZ risk factors may be similar to symptoms of other neurodevelopmental psychi-atric disorders, transition processes from child & adolescent to adult systems of care should include both high risk people as well as subject with other neurodevelopmental psychiatric disorders with different levels of severity. This descriptive mini-review discuss the need of innovative clinical approaches, re-considering specific diagnostic categories, stimulating a careful analysis of risk factors and promoting the appropriate use of new and safer medications.

## 1. Introduction

Understanding the pathogenesis of schizophrenia has dramatically changed in the last four decades. Throughout most of last century, the prevailing biological perspective was based the idea that schizophrenia (SCZ) results from pathological processes occurring in early adult life, shortly before the onset of symptoms In a dramatic break with this tradi-tional formulation, the advent of modern neuroimaging, the progressively refining of ge-netic research and, more in general, the developmental neuroscience, led to the current conceptualization of SCZ as a neurodevelopmental disorder starting before birth, present-ing a large variety premorbid and prodromal symptoms leading to progressive impairment for the patient and, in relatively minority of cases, resulting in the full blown syndrome [1,2,3]. It is important to recognize that many neurobiological processes leading to SCZ are common to other neurodevelopmental disorders, such as Autism Spectrum Disorder (ASD), Attention Deficit Hyperactivity Disorder (ADHD), Intellectual Disability (ID) [4]. This de-gree of complexity is parallel to a profound reconceptualization of the general psycho-pathology and, as a consequence, of the systems of care organization. Far from considering children and adolescents as “small adults” suffering from unusual, severe variants of adult disorders, adult patients are rather currently considered as children with specific predispo-sitions who missed appropriate support during development.

For all these reasons, transitional psychiatry from childhood to adulthood has be-come an important paradigm in psychiatry, and schizophrenia may be an example of this approach, describing specific neurobiological underpinnings, clinical presentations, re-sponse to clinical interventions throughout the development, in order to justify the need of a close integration and continuity between clinical services for children and adolescents and those for adults [5]

## 2. Genetic and Epigenetic in SCZ

The underpinning genetic architecture of SCZ remains unclear. In a seminal genome-wide association study (GWAS) of common variants in 36,989 schizophrenia cases and 113,075 controls, 108 independent loci were found significantly associated with schizophrenia status [6].While each significantly associated loci was found to confer only a small increase in risk (i.e., median odds ratio (OR) per SNP = 1.08), when the effects of all nominally associated (*p* < 0.05) loci were considered together as a single polygenic risk score (PRS), SCZ PRS was able to explain 18.4% of the variance in case versus control status. Specifically, 40% of the 108 associated loci were located within the sequence boundaries of a single protein coding gene, the remaining associated SNPs were located in non-protein coding regions of the genome, suggesting that many common variants associated with Adult Onset Schizophrenia (AOS) are likely to contribute to disease risk by altering the level of expression of specific proteins, rather than more directly altering protein structure. Implicated genes at associated loci included DRD2, (dopamine D2 receptor), gltatergic signaling and plasticity, (GRIA1, GRIN2A, and SRR); and genes encoding 68 voltage-gated calcium channels, (CAC-NA1C and CACNB2) [6].

A more recent meta-analytic GWAS of common variants in schizophrenia included an additional 5220 schizophrenia cases and 18,823 controls and identified 145 independent loci significantly associated with schizophrenia [7]. Schizophrenia-associated SNPs were enriched for genes that are intolerant to mutation, genes involved in synaptic transmission, and genes that are targets of the fragile X mental retardation protein (FMRP), which is known to regulate the protein-level expression of genes involved in brain development and synaptic plasticity [8]. Together, these seminal studies provided compelling evidence that common risk variants for schizophrenia converge onto neuronal and synaptic gene-sets. SCZ patients have also been found to carry rare and ultra-rare deleterious mutations [9,10]: although the effect size of this overall increased burden is relatively small (OR = 1.07), aggregated at the gene-set lev-el, rare deleterious mutations in schizophrenia patients were found to be enriched for genes that are intolerant to mutation, genes that are expressed specifically in neurons, gene targets of FMRP [11], and genes that are components of synaptic gene-sets, such as the N-methyl-D-aspartate receptor (NMDAR) and activity-regulated cytoskeleton-associated protein (Arc) complexes, all critically involved in modulating synaptic plasticity [9,10,12]. Copy number variants (CNVs) are a particular class of structural variants in which segments of DNA are deleted or duplicated, resulting in genomic imbalances in the normal number of copies of DNA in the region. Large (e.g., >100 kb), rare CNVs (i.e., observed in <1% of the population) have been consistently associated with schizophrenia (e.g., OR = 1.15) [13,14,15]; and have yielded important insights into the genetic etiology schizophrenia, CNV loci associated with schizophrenia, include deletions at the 22q11.2, 2p16.3 (NRXN1), 3q29, 15q11.2, and 15q13.3 loci, duplications at the 16p11.2 and 7q11.23 loci, and deletions or duplications at the 1q21.1 and 7p36.3 (VIPR2) loci [16,17,18].

About 2.5% of patients with SCZ are estimated to carry CNVs at one or more schizophrenia-associated loci [17]. Interestingly, CNVs at many of these specific loci are de novo (not present in parents) and have pleiotropic effects, as they are also associated with broader neurodevelopmental disorders such as ASD and ID [17,19,20,21]. Similarly to common variants associated with SCZ, as well as other rare and de novo variants, SCZ-associated CNVs disproportionately affect neuronal and synaptic gene-sets [22], including components of the postsynaptic density, and NMDAR and Arc complexes [19], and sets of genes that are involved in excitatory and inhibitory [23].

Growing evidence indicates common and rare variants interact to increase risk. Thus, the total burden of common schizophrenia-associated risk alleles that a given individual carries can be summarized by their schizophrenia PRS, which is calculated as their weighted sum of schizophrenia risk-associated SNP alleles at GWAS studies [24]. While SCZ patients have higher PRS than controls, regardless of CNV carrier status, patients who carry risk CNVs that have been previously associated with SCZ have lower PRS compared to patients without risk CNVs [25,26].

There are relatively few genetic studies for Childhood Onset Schizophrenia (COS). Preliminary evidence suggests that in addition to sharing genetic risk factors with Adult-onset Schizophrenia (AOS), the genetic architecture of COS may include greater loading from variants that also confer risk for other neurodevelopmental disorders, such as ASD, ID, and epilepsy: in a study of 130 COS probands and 103 of their healthy siblings, COS probands were found to have significantly higher schizophrenia PRS than their siblings, as well as higher polygenic risk for ASD [27]. Elevated rates of large CNVs have also been found in COS [18], including in CNVs associated with SCZ and other neurodevelopmental disorders. Rates of large, rare CNVs appear to be higher in COS patients compared to controls as well as to patients with AOS; 11.9% of COS probands were estimated to have a neurodevelopmental disease-associated CNV compared to 1.5% of their healthy siblings and 1.4–4.9% of AOS patients [27,28,29]. In particular, a high number of COS probands have been found to carry CNVs at the well-known 22q11.2 locus, which is known to increase risk for multiple psychiatric and developmental disorders, including schizophrenia, ASD, ID and ADHD [27,28,29].

Recently, genome research focused on epigenetic. This mechanism has a role in regulating brain functions, neurogenesis, neurodegeneration, neuronal activity, and cognition. Epigenetic mechanism induces hereditable changing in phenotype, influencing genome functions through modification in DNA, histone, and chromatin structure [30]. The main epigenetic mechanisms consist in DNA methylation, post-translational histone modification, and RNA interference. The totality of these modifications define epigenome, impacting in gene expression program in a temporally and dynamic way [31]. According with neurotransmission hypothesis of SCZ, some authors examined dopaminergic pahtway. They found a hypermethylation in dopamine regulation: DRD1-5 and COMT which encode for dopamine degrading enzymes; DRD4 promoter in peripheral blood. Hypomethylation has been reported for DRD2, DRD4 and DRD6. In the GABAergic pathway, hypermethylation at the pro-moter regions has been found in RELN and GAD1: the most representative genes of the inhibitory neuro-signaling system [32]. Hypermethylation in BDNF I promoter has been also associated to SCZ [33]. Other methylations were detected in genetic loci involved in the regulation of embryonic development such as SOX10 [34] and BAIAP2 [35] in the brain, responsible of dendritic spine density [34]. Epigenetic mechanism act also on immune function with methylation in CTLA4a and OXTR [36]. OXTR encode for the oxytocin receptor which in known to be linked to social cognition deficit in SCZ [37]. Genome methylations has been described in GRIA1: an ionotropic AMPA receptor subunit, important for synaptic plasticity [38,39]. Actually, there are not consensus on the meaning of histone modifications, but some authors investigated H3K9 di-methylation as a putative epigenetic factor underlining SCZ pathogenesis. Micro RNA dysregulation is another epigenetic mechanism probably involved in synaptogenesis that is altered in SCZ [40].

In the epigenome paradigm, age is an important factor because during early developmental phases, the organism is more sensitive to chemical and environmental influences [32]. For these reasons, prenatal and perinatal factors should be particularly considered in clinic for their epigenetic potentiality [33,34]. During pregnancy, maternal infections and maternal immune activation, diet, toxic factors (including alcohol and substance use) has been associated to an increase risk in develop SCZ [35]; asphyxia, maternal and paternal age, low weight at birth are also important information. In addition, in the first years of the life, environmental conditions seem to be associated to SCZ probably for epigenetic influences. In postnatal period, repeated experience of trauma, neglect, substance use (especially cannabis and stimulant use such as cocaine, methamphetamines etc.) are the most important epigenetic risk factor for SCZ [36].

## 3. EOS as a Neurodevelopmental Disorder: Neuroanatomic Evidence

Since 1990, children and adolescents with COS and their full siblings participated in a National Institute of Mental Health (NIMH) prospective longitudinal study including brain imaging, genetic, and treatment studies [37]. These morphometric studies with brain imaging have provided novel insights into brain development in children with COS, showing global gray matter (GM) loss with ventricular expansion [38], with a parieto-frontal-temporal pattern of spread during adolescence [39], interpreted as an exaggeration of the normal GM developmental pattern [40], who became after age 20 less marked and more circumscribed to the prefrontal and superior temporal cortices [41]. Consistently with a neurodevelopmental model of SCZ, and with genetic factors of predisposition, as above reported, parallel studies in healthy full siblings of probands with COS showed initial cortical GM deficits (most prominent in prefrontal and temporal cortices, but minimal parietal GM loss, only at much younger ages), without a different progression during adolescence, compared to proband siblings and with normalization after age 20 [41].

These GM abnormalities may be, at least in part, familial markers, or endophenotypes, indicating dysregulation of the patterns of GM development, more pronounced in earlier periods of development, around the real psychosis onset, shared by both probands and siblings, with healthy siblings presenting protective/restitutive factors [42,43]. These findings suggest a dynamic expression a diffuse interconnected abnormality in the development of many cortical circuits [44], with overall similarities in the patterns of GM development in probands and healthy siblings, which vary over time, with a more pronounced expression in the first phases of the illness and/or at an earlier age, probably reflecting a stronger genetic vulnerability interacting with the early dysregulated neurodevelopment, probably the crucial deficit, later influenced by environmental factors, including treatments [45].

The absence of parietal deficits in healthy siblings may also suggest a two-hit model, in which posterior parietal brain regions appear vulnerable to environmental triggers (traumas, substance use, infections, neglect, social isolation, etc.), while frontal regions may be more related to a genetic liability, with resulting deficits in executive functions.

Interestingly, children with pediatric Bipolar Disorder (BD) type 1 (with psychosis) (a “neurodevelopmental” form of BD) also present developmental trajectories of cortical GM alterations, with a more subtle and distinct pattern of cortical GM gain in left temporal cortex and loss in right temporal and bilateral subgenual cingulate cortices [42,43], supporting a specificity of the GM findings in the neurodevelopment of SCZ and BD.

## 4. Overlaps between Early-Onset SCZ, ASD and Other Neurodevelopmental Disorders

High rates of co-occurring neurodevelopmental disorders, predating the onset of VEOS, support the notion of a pan-developmental disturbance consisting of impairments in neuromotor, receptive language, social and cognitive development [46]. In the NIMH sample, at least 2/3 of children with COSpresented premorbid developmental disorders [47], and at least 1/4 have met criteria for ASD before the onset of psychotic symptoms [48], possibly representing a premorbid phenotype of early-onset SCZ.

Phenomenological, genetic, environmental, and imaging evidence supports the overlap between ASD and SCZ, highlighting similarities and areas of distinction [49]. According to Chisholm and colleagues, four models can be hypothesized, not mutually exclusive, as different subgroups of patients may be interpreted according one or other model: (1) the increased vulnerability model (ASD are more at risk of psychosis due to their ASD, but the conditions are separate); (2) the diametrical model (ASD and psychosis are opposite ends of a continuum of overlapping constructs); (3) associated liabilities model (factors that increase risk of one condition also increase risk of the other, but they remain separate); (4) multiple overlapping etiologies model (some factors that lead to developing ASD also lead to developing psychosis, but others do not, leading to distinct but often similar or overlapping presentations).

Developmental pathways from ASD to schizotypal traits are possible precursors of the transition to SCZ [50,51,52]. In the vulnerability to psychosis in ASD patients both emotional dysregulation [53] and ADHD [52,54] have been explored. Emotional dysregulation, interpreted as a neurodevelopmental disorder of emotion regulation, in patients with ASD was positively and significantly associated with positive and disorganized schizotypal traits (as a proxy for psychosis-proneness), suggesting that this developmental feture may relate to the development of schizotypal traits and psychosis in autism, at least in a subgroup of patients with early-onset SCZ [55]. In children with ASD and later very early-onset SCZ, the association between ASD and ADHD was markedly associated with more severe disorganized thought, disorganized behavior, and negative symptoms of SCZ [52,54]. These findings support not only an interrelation between ASD and SCZ, but also the differential influence of other neurodevelopmental syndromes on psychotic phenotype, and themultidimensionality of SCZ spectrum in children with ASD [52].

## 5. Prodromal Symptoms

The previous sections underline that SCZ begins early, with genetic vulnerabilities and early neuroanatomical abnormalities. Thus, in recent years, researchers and clinicians have focused on identifying and managing, at a clinical level, the early phases of the disorder to prevent or delay its onset. Indeed, previous research highlighted that the early identification of disorder may improve outcome [56,57].

The most commonly used protocol for assessing help-seeking individuals suspected to be in an initial prodromal phase of psychotic disorders, specifically SCZ, has been the “Ultra-High Risk” (UHR) approach [56]. Criteria for identifying help-seeking individuals at risk for transition to psychosis (Ultra-High Risk—UHR) include: (1) Attenuated Psychotic Symptoms with subthreshold positive psychotic symptoms (APS); (2) Brief Limited and Intermittent Psychotic Symptoms (BLIPS—brief psychotic episode of less than 1 week’s duration that spontaneously remits without antipsychotic medications); (3) genetic risk (schizotypal personality disorder or a first-degree relative with psychosis) combined with a significant impairment in the global functioning: the so call Genetic Risk and Deterioration factor (GRD) [58,59,60]

Recently, based on these criteria, DSM -5 included the Attenuated Psychosis Syndrome (APS) in Section III—“Conditions for further study” [61]. The diagnosis of APS requires the presence of delusions, hallucinations, or disorganized speech in an attenuated form that are present at least once per week for the past month, not better explained by another psychiatric or medical diagnosis, and which have never been severe enough for the individual to meet diagnostic criteria for a psychotic disorder [61]. Studies on the clinical profile of children and adolescents with APS are few. Tor et al. [62], in a systematic review of literature, showed that children and adolescents with APS have high levels of attenuated positive symptoms, specifically perceptual abnormalities and suspiciousness, and comorbid non-psychotic psychiatric disorders, such as depressive and anxiety disorders. In addition, compared to healthy controls, children and adolescents with APS showed lower general intelligence. Doltz et al. [63], compared a sample of children and adolescents with APS with a healthy control group. The results showed that 79.1% of the children and adolescents with APS reported suspiciousness and delusional ideas. Additionally, they presented psychiatric comorbidities associated with APS (65.9%), including a prevalent diagnosis of depressive disorder (60%). Moreover, 49.5% of the children and adolescents with APS had a first- or second-degree psychotic relative who met a significant number of the UHR criteria [63]. Finally, children and adolescents with APS showed more neurodevelopmental impairment (e.g., lower gestational age) compared to healthy control group. Overall, children and adolescents with APS presented a pattern of neurodevelopmental impairment and clinical complexity similar to patients with SCZ spectrum disorders; these results support the neurodevelopmental hypothesis and highlight the need of longitudinal studies focusing on the assessment of other clinical markers of SCZ in children and adolescents with APS specifically. Markers to consider could be the presence of abnormalities in developmental domains (psychomotor, language, and reading/writing skills), the presence of childhood trauma or an early diagnosis of non-psychotic psychiatric disorder like Social Anxiety Disorder or Obsessive-Compulsive Disorder [64,65].

In the clinical approach, another useful and important concept is the “basic symptoms” paradigm which has its neurobiological correlate in the dysfunctions of mesolimbic system and prefrontal cortex [66]. The basic symptoms are subclinical disturbances of the subjective experience that precede clear delusions and hallucinations. The conversion to psychosis can be determinate by everyday events that overstrain coping capability [66,67,68]. This prodromal phase can be closely explored, focusing on the dimension of self-awareness with the EASE (Examination of the Abnormal Experience of Self) interview [68] while basic symptoms can be evaluated by a semi structured interview, the SPI-A (Schizophrenia Proneness Interview-Adult) [69].

## 6. Clinical Features

According to DSM-5, Very Early Onset Schizophrenia (VEOS) is characterized by an onset of psychotic symptoms before 13 years, while Early Onset Schizophrenia (EOS) presents an onset between 13 and 18 years [70,71]. Estimated prevalence rates of EOS and VEOS are respectively 1–2/100 and 1/10,000 occurs between 13 and 17 years [72]. SCZ onset is rare in developmental age: EOS and VEOS have respectfully a prevalence of 1–2/100 and 1/10,000 [73].

Prominent features of VEOS are premorbid developmental (cognitive, language, motor), disabilities, socio-communicative disturbances. The insidious onset, in frequent continuity with previous developmental delays, as well as the hesitancy on the part of clinicians to make a diagnosis of schizophrenia in a child usually delay the recognition of the syndrome. Elementary auditory hallucinations are the most frequent positive symptom, while visual and tactile hallucinations are rarer. Delusions are less complex than in adolescents and are usually related to childhood themes. Negative symptoms are largely predominant, namely flat or inappropriate affect. A severe deterioration from the previous level of functioning is present in all these children, with poor outcome [74]. A long duration of untreated psychosis (DUP), compared to EOS, due to the difficult diagnosis and the hesitation in starting pharmacotherapy, further worsens the risk of negative prognosis [75]. The DUP for EOS lasts until 5–3 times more than the duration of untreated psychosis of adult SCZ [76].

A metanalysis on child and adolescent psychosis, reporting on clinical characteristics, diagnostic trajectories, and predictors of illness severity and outcomes, based on 35 studies covering 28 independent samples and 1506 patients, showed that the most frequent psychotic symptoms were auditory hallucinations (81.9%), delusions (77.5%; mainly persecutory [48.5%]), thought disorder (65.5%), bizarre/disorganized behavior (52.8%), and, significantly, flat or blunted affect/negative symptoms (52.3%). Comorbidity was frequent, particularly posttraumatic stress disorder (34.3%), ADHD and/or disruptive behavior disorders (33.5%), and substance abuse/dependence (32.0%). Longer duration DUP and poorer premorbid adjustment predicted less improvement. Five studies directly comparing early-onset with adult-onset psychosis found longer DUP in EOP samples (18.7 ± 6.2 vs. 5.4 ± 3.1 months) [77].

Autistic traits correlate with higher rates of depression and suicide [11]. Substance use is common in adolescent-onset SCZ, and it strongly interacts with genetic and early epigenetic factors [37]. [78]. Moreover, early-onset substance use during adolescence damages myelination and pruning [78]. During lifetime, three quarters of SCZs have issues with substance and alcohol, compared to 16% in controls [79]. The 50% of SCZs use substances during the first episode [80]. Comorbidity between SCZ and substance use is associated with less treatment adherence, worse outcome, high hospitalization rates, violence, and suicide [81,82]. Apathy, anhedonia, abulia, dysthymia and other symptoms also framed in SCZ basic symptoms may be at first alleviated by substances, and then worsened by chronic use and addiction [83].

One of the most impaired psychopathological domains in SCZ is neurocognition [84,85]. Since Bleuler [86], cognitive impairment in SCZ has been focused on associative disturbance. Later, difficulties in abstract attitude and overinclusive thinking were added to the core features for SCZ [87]. Currently, a huge amount of evidences has found deficits in speed of processing, attention/vigilance, working memory, verbal/visual learning, problem solving, and social cognition [88]). In SCZ, attenuated cognition alterations are present since the age of five years [89]. These deficits can be better explained and understood exploring the overlaps between SCZ and other neurodevelopmental disorders, such as Autism Spectrum Disorders (ASD) and Attention Deficit Hyperactivity Disorder (ADHD).

## 7. Pharmacological Treatments in Early Onset Schizophrenia

Although VEOS and EOS are rare compared to the adult onset SCZ, and differ in some clinical features, diagnostic criteria are grossly the same, and allow timely diagnosis and treatment, with positive implications on short- and long-term prognosis. Close monitoring of UHR subjects can further increase the implementation of effective treatments, allowing us for an early recognition and more timely treatments in those who convert to fully developed psychosis (who should be the only to receive a pharmacological treatment) [76].

Even though psychoeducational interventions, including parent support, social interventions, cognitive remediation, and psychotherapy are crucial, namely during the prodromal phase and after the acute phase [90], pharmacological intervention is mandatory when psychotic symptoms lead to a diagnosis of SCZ, although sometimes prescription difficulties may delay a prompt treatment [91]. An early psychopharmacotherapy may improve the SCZ underlying impairment, possibly carrying out a neuroprotective action [92]. Antipsychotic medications present an effect size of 0.5 compared to a mean effect size of psychiatric medications of 0.3–1.0 [93].

Second Generation Antipsychotics (SGAs) are among the psychotropic drugs with the most relevant increase rate of prescription all over the world [94]. They are more often used in not psychotic conditions such as disruptive behavior disorders, mood disorders, ASD and behavior disorders in people with ID [95]. A metanalysis on 32 studies on SGAs use in children and adolescents for non-psychotic conditions, shows a greater efficacy of SGAs, measured in Number Needed to Treat (NNT), on manic symptoms, extreme mood lability, irritability, disruption and aggression than on psychotic symptoms in SCZ (NNT 2–5 in non-psychotic condition in comparison with NNT 3–10 for SCZ) [95]. As an example risperidone in autism has an effect size of 1.1 with an NNT 1.610 mainly on irritability and aggressiveness but 0.4 on language and social withdrawal [96]; in schizophrenia, effect size for risperidone is 0.62 [97], for aripiprazole 0.42 [98].

A network meta-analysis including all the antipsychotics used in children and adolescents with SCZ showed that all the included medications, except ziprasidone, are efficacious compared to placebo in overall schizophrenic symptoms, and clozapine is the most effective [99]. A cohort of 131 patients VEOS observed in 24 months treated with clozapine show that 72% of the sample has a good response, suggesting a better response in VEOS than in adult SCZ [100]. These data supports clozapine as an option in VEOS, especially in the refractory forms, consistently with data from adult populations.

Affective symptoms sometimes co-occur with psychotic symptoms, like in schizoaffective disorders. SCZ and BD seem to be converging in many psychopathological domains and genetic liability [101]. For these reasons, affective symptoms should influences pharmacotherapy. Lurasidone and Olanzapine (combined with Fluoxetine) are the only drugs approved by FDA (but not by EMA) for bipolar depression in children and adolescents [102,103]. Clozapine was the only antipsychotic associated with decreased risk of attempted or completed suicide among adult patients with SCZ, and it should be considered as first-line treatment for this type of high-risk patients [104].

Lithium is the most effective anti-suicidal medication, and the retard formulation may be more suited for adolescent patients [105,106,107].

## 8. Safety Antipsychotics in Youth

The SGA are similar each to another in efficacy profile in children and adolescents, but they significantly differ in safety profile and tolerability [95], mainly regarding metabolic side effects and hyperprolactinemia [108], even if cardiac side effects, extrapyramidal symptoms (EPS) and akathisia should also be closely monitored [109].

Metabolic effects and hyperprolactinemia are common and important side effects of antipsychotics [110]. Metabolic syndrome is a risk factor for morbidity and mortality from cardiovascular diseases, and also negatively influence quality of life [111]. SCZ patients are more prone to metabolic syndromes and require regular monitoring, dietary advices, lifestyle changes [112]. Hyperprolactinemia induced by antipsychotic drugs interferes with endocrine system with possible consequent side effects such as menstrual disturbances, galactorrhea, sexual dysfunction, gynecomastia, infertility; in the long-time decreased bone mineral density, and breast cancer has been seen too [113]. According with the previously mentioned network meta-analysis [99], regarding weight gain and prolactin, lurasidone, molindone, and ziprasidone resulted better tolerated drugs. The highest weight gain has been registered by clozapine, quetiapine, and olanzapine [104], while risperidone, haloperidol, paliperidone and olanzapine were associated with high prolactin levels [114]. Lurasidone presents the lowest weight gain, moderate hyperprolactinemia, and the best metabolic profile [115].

Cardiac prolongation of the correct QT interval of the Electrocardiogram is an indirect sign of torsade de pointes risk, a condition that might lead to sudden cardiac death [116]. Although in childhood and adolescence, pathological prolonged QT interval, associated to antipsychotic treatment, is not frequently reported [117], quantity and quality of evidence are lower than in adulthood. Thus, a careful monitoring is recommended [118].

The EPS are usually associated with high potency First Generation Antipsychotics (FGA) [119]. According to Rybakowsky, in patients treated for a first episode of psychosis 1% have dyskinesia, 1,8% dystonia, 11% parkinsonism, and up to 10% present akathisia [120]. In pediatric samples clozapine and quetiapine showed lower rates of EPS [93], while risperidone, olanzapine and aripiprazole have a non-negligible incidence of acute EPS [109,121]. Acute dystonia may appear after few days by starting or increasing antipsychotic dose, although studies indicate SGAs have a low relative risk (RR: 0.19) for acute dystonia when compared with FGA [122].

Neuroleptic Malignant Syndrome (NMS) is a rare but potentially fatal side effect. It is more likely in young male patients and in FGA prescription. Dehydration and agitation are risk factors. It is characterized by fever, stiffness, diaphoresis, fluctuation of the level of consciousness, autonomic dysfunction, elevation of creatine kinases, leukocytosis, alteration of liver function tests and renal function tests. Management of NMS requires stopping antipsychotic drug, rehydration, dantrolene administration and sometimes ventilation in a medical or A&E unit [123,124].

Another antipsychotic side effect is the antipsychotic- induced dopamine supersensitivity psychosis (SP), when a persistent blockade of D2 receptors lead to an upregulation of D2 receptors [125] This is a frequent and underestimate phenomenon, occurring in the 30% of schizophrenic patient and in the 70% of resistant forms of SCZ [126], namely in prolonged treatments or after a discontinuation. It is worthy to note SP may be the hidden reason of many refractory SCZ: clinical consequences are psychotic relapses that requires higher doses of antipsychotics and tardive dyskinesia. A good strategy to prevent a persistent D2 receptor block and the consequent SP should be evaluating movement disorders and prolactin level to regulate dose of antipsychotic since EPS and hyperprolactinemia are present just when D2 receptor are occupied at least for a 78% [126].

## 9. A Transitional Model for SCZ

The re-conceptualization of SCZ as neurodevelopmental disorders with fetal origin [127], and the importance of gene x environment interaction significantly before the onset of clinical symptoms [128] should have implications not only in terms of nosological categorization and clinical management, but also in terms of organization of the systems of care, particularly in the transition from adolescent to adulthood.

In most of European countries, medical care for children and adolescent with psychiatric disorders is independent and splitted from that for adults. To this different organization correspond specific training (either as independent curriculum or as sub-specialty of adult psychiatry training), and specific Scientific Societies affiliation. The construct of developmental psychopathology (so far a relatively unusual construct for adult psychiatrists) may be the cultural basis to overcome this split, and its negative consequences in patients management [129,130]. Both the lack of timely interventions as well as inappropriate intervention (i.e., obesity induced by long term use of SGA) may lead to severe impairment later in life. In the last two decades a five-fold increase in the use of antipsychotic medication in adolescents, and 9-fold increase in children (compared to the 2-time increase in adults), usually prescribed for non-psychotic disorders [94], indicate the different evolving attitude of psychiatrists in the use of medications, with poor integration between child/adolescent and adult psychiatrists. These new attitudes are supporting innovative approach for a neuroscience based nomenclature in the classification of drugs [131]. The increased knowledge that “antipsychotic” medication should be used to modulate salience of both external stimuli and internal feeling [132], rather than just managing psychotic symptoms, may be considered an important contribution of child & adolescent psychiatrists to the systems of care for mental disorders.

The difficult communication between mental health services in the transition of patients from adolescence to the adulthood is also based on the poor expertise of adult psychiatrists in neurodevelopmental psychiatric disorders, mainly ASD, ADHD and ID (i.e., adult psychiatry ignores important anamnestic elements), patients are not adequately supported in adult service, with not always justified change of diagnosis and treatment.

Another crucial issue is the management of adolescent patients at risk for psychosis. As mentioned in a previous section of the paper, only 30% of young people seeking help for psychotic-like symptoms may considered at high risk of psychosis, and only less than 20% of these high risk subjects eventually develop clear psychotic symptoms, but about 70% of them may result functionally impaired on global functioning often developing depression, anxiety, substance abuse, bipolar disorder or persisting attenuated psychotic symptoms [133]. A recent study including 2330 youth to early intervention services were assessed longitudinally, 4.3% (n = 100) met criteria for new-onset full-threshold Bipolar Disorder (FT BD) and 2.2% (n = 51) met criteria for a new-onset FT PD. The emergence of FT BD was associated with older age, lower social and occupational functioning, mania-like experiences (MLE), suicide attempts, reduced incidence of physical illness, childhood-onset depression, and childhood-onset anxiety. The emergence of a PD was associated with older age, male sex, psychosis-like experiences (PLE), suicide attempts, stimulant use, and childhood-onset depression [134].

Considering that many risk factors may be similar to symptoms of neurodevelopmental psychiatric disorders, transition processes should include both high risk people as well as subject with neurodevelopmental psychiatric disorders with different levels of severity.

Two main models of transition between child/adolescent and adult services may be considered: using a “transition team” that operates independently from both services to bridge the gap, or the use of shared care official protocols interlocking child/adolescent and adult services facilitating a gradual transfer of care. The independent transition service model has been implemented in early intervention in psychosis, but with inconsistent effects [135,136]. The main feebleness of this model is the introduction of additional and unnecessary splits within the system. The interlocking model may work when it is flexible to the needs of young adults rather than focused specifically on chronological age.

According to the interlocking model transition should be planned by the services of first referral. During transition, child/adolescent and adult services, both at community level and at a second-or third-level hospital, should consider meeting and shared full information, considering evidence-based, up-to-date recommendations about the diagnosis and management of psychiatric disorders at different developmental stages as part of their continuing professional development. Appropriate adult services should include primary care, adult community mental health teams and access to dedicated services for specific disorders.

Transition protocols should be available to all clinical teams and should include psychoeducational material that provides high quality, comprehensive, and appropriately written information for both young people and their parents/caregivers. This material should include information about how management of their own symptoms and problems, and access advice and support. Information should also be developed in a media format that is readily accessed by young people, e.g., use of phone applications and internet sites.

Full information about adult psychiatric and general psychiatry services should be made available to the young person and their family. Full information about the previous young person care should be available to the adult teams, including a detailed clinical transition report. Collaboration with educational and/or occupational agencies is usually also needed.

## 10. Conclusions

Translational psychiatry proposes a new psychopathological paradigm in SCZ [5]. Firstly an important acquisition is to consider SCZ as a neurodevelopmental disease [1]. SCZ spectrum is a condition in which genes and environment interact in different phases of the development, causing an individual neurobiological vulnerability [71]. Continuous distress may lead to transdiagnostic conditions as emotional dysregulation [137], SCZ basic symptoms [68], psychosis [67]. An early and timely diagnosis and treatment is mandatory [138], too watchful and waiting conservative approach may risk to increase DUP and worsen prognosis and outcome in some cases [76,139].

In this context translational psychiatry may change psychiatrist clinical approach reconsidering old categories, stimulating a careful analysis of risk factors, and promoting the correct use of new and safer molecules.

## 11. Legend

Adult Onset Schizophrenia (AOS); Autism Spectrum Disorders (ASD); Attenuated Psychotic Symptoms (APS); Attention Deficit Hyperactivity Disorder (ADHD); Bipolar Disorder (BD); Brief Limited and Intermittent Psychotic Symptoms (BLIPS); Childhood Onset Schizophrenia (COS); Central Nervous System (CNS); Copy number variants (CNVs); Diagnostic and Statistical Manual of Mental Disorders—5th edition (DSM -5); Duration of Untreated Psychosis(DUP); Early Onset Schizophrenia (EOS); EASE (Examination of the Abnormal Experience of Self); extrapyramidal symptoms (EPS); First Generation Antipsychotic (FGA); Food Drug Administration (FDA); Fragile X Mental Retardation protein (FMRP); Full-Threshold Bipolar Disorder (FT BD); Full-Threshold Psychotic Disorder (FT PD); Genetic Risk and Deterioration factor (GRD); Genome-Wide Association Study (GWAS); Gray Matter (GM); Intellective Disability (ID); Mania-Like Experiences (MLE); National Institute of Mental Health (NIMH); Neuroleptic Malignant Syndrome (NMS); N-methyl-D-asparate receptor (NMDAR); Number Needed to Treat (NNT); Polygenic Risk Score (PRS); Psychosis-Like Experiences (PLE); Schizophrenia (SCZ); Second Generation Antipsychotics (SGAs); SPI-A (Schizophrenia Proneness Interview-Adult); Supersensitivity Psychosis (SP); Ultra-High Risk (UHR); Very Early Onset Schizophrenia (VEOS).

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
