# Peer review of "A Neurodevelopment Approach for a Transitional Model of Early Onset Schizophrenia"

_brainsci, 2021, doi:10.3390/brainsci11020275_

Round 1

Reviewer 1 Report

The  paper “Translational Psychiatry in Schizophrenia” written by De Berardis et al., considers translational aspects of schizophrenia. The review is interesting and the paper is well written. However, there are some concerns related to the paper.

  1. Point 2. p. 2: It is not clear which of the risk factors have epigenetic background.
  2. Point 4. p.3: Schizophrenia is also heritable and such information should be included.
  3. Point 8. p.8, line 360: What kind of stimulants did authors refer to.
  4. General: There are a lot of abbreviations through the manuscript. I would suggest to add a list of used abbreviations to improve reading and understanding of the manuscript.

Reviewer 2 Report

This is a well written mini review. The efficacy of antipsychotics in Very Early Onset Schizophrenia, Early Onset Schizophrenia, and prodromal symptoms are well addressed in the review.  However, side effects of some antipsychotics including insidious psychosis is not discussed.  It would be appropriate if the authors could address these issues in regards to targeting, diagnosis and treatments of schizophrenia in young people. 
